# Mindfulness Meditation as Adjunctive Therapy to Improve the Glycemic Care and Quality of Life in Patients with Type 1 Diabetes

**DOI:** 10.3390/medsci9020033

**Published:** 2021-05-21

**Authors:** Rishi Shukla, Manisha Gupta, Neha Agarwal, Anurag Bajpai

**Affiliations:** 1Society for Prevention And Awareness of Diabetes, Kanpur 208002, Uttar Pradesh, India; drrishishukla@gmail.com (R.S.); doctormanishaknpr@gmail.com (M.G.); 2Regency Center for Diabetes Endocrinology & Research, Department of Endocrinology, Kanpur 208002, Uttar Pradesh, India; dranurag.bajpai@gmail.com

**Keywords:** Mindfulness Meditation, type 1 diabetes, quality of life

## Abstract

Background: Mindfulness Meditation (MM) is known to improve glycemic control and enhance the quality of life (QoL) in type 2 diabetes (T2D) patients. Unfortunately, the role of meditation in type 1 diabetes (T1D) has not been studied extensively. Therefore, we conducted this study to determine the effect of MM on the glycemic control and QoL in people living with T1D. Methodology: Thirty-two adults living with T1D were equally randomized into intervention (meditation) and control groups. The glycemic control and QoL were assessed at the baseline and after six months of intervention. Results: A total of thirty-two adults (15 males, 46.9%; mean age 23.8 ± 6.6 years) with type 1 diabetes (mean diabetes duration 12.7 ± 6.2 years) participated in the study. At the end of six months, a statistically significant improvement was seen in the mean blood glucose level in the control group (222.4 ± 77.8 versus 182.6 ± 52.0; *p* = 0.007) and the intervention group (215.3 ± 50.1 versus 193.2 ± 31.8; *p* = 0.008). Additionally, there was a significant reduction in the total diabetes distress score in the intervention group (1.6 ± 0.3 versus 1.3 ± 0.3; *p* = 0.003), while no change was observed in the control group (1.6 ± 0.7 versus 1.7 ± 0.4; *p* = 0.762). A statistically significant improvement was noticed in the health and functioning domain in the intervention group (*p* = 0.023). Conclusions: An improvement in the glycemic control and quality of life of the patients was observed in our study. MM certainly plays an important role in attaining peace of mind and helps patients to channel their energy in a positive direction.

## 1. Introduction

Type 1 diabetes (T1D) is the most commonly diagnosed endocrine disorder across the globe. Individuals living with T1D are at an increased risk of psychological stress, either due to the underlying disease or to the complexities involved in the management of diabetes [1]. While positive stress can lead to motivation and appropriate action, negative psychological stress or distress can have a significant impact on the neuroendocrine, cardiovascular, immune and central nervous systems [2]. Indeed, there is a high prevalence of anxiety and depression among diabetic individuals. Although, depression tends to be severe but is often ignored, unrecognized or unaddressed. Diabetes and its comorbidities are associated with significant diabetes distress, with a possible impact on the glycemic parameters and quality of life (QoL).

Nonmedicinal modalities such as diet management, physical activity and stress management are well-known adjuvant therapies in the management of type 2 diabetes (T2D) [3]. Meditation is known to reduce fasting, postprandial blood sugar and HbA1C. Mindfulness Meditation (MM) is a known tool to reduce restlessness and enhance the QoL in T2DM [4].

Unfortunately, the role of Mindfulness Meditation has not been studied widely in individuals living with T1D. We believe that stress reduction is likely to improve the glycemic control and QoL in T1D patients. Therefore, we conducted this study to determine the effect of MM on the glycemic control and quality of life (QoL) in people living with T1D.

## 2. Materials and Methods

This study was conducted in the Endocrine Outpatient Department of a tertiary care hospital in North India (Regency Healthcare, Kanpur, Uttar Pradesh, India). It was approved by the institutional ethics committee (approval code: RHL-IEC-16002). Patients suffering from diabetes-related complications such as proliferative retinopathy, autonomic neuropathy, diabetic nephropathy, established or recent onset coronary heart diseases and pregnancy were excluded. Thirty-two adults with T1D were randomized (by the chit method) after informed consent into the intervention (meditation) and control groups. Participants in the intervention group were formally educated about MM by a skilled meditation coach. The training program was divided into two components, breathing exercises and meditation (Appendix A). They were advised to practice MM for at least 20 min daily. Compliance and adherence were ensured by weekly telephonic calls and in-person monthly follow-ups.

For the glycemic control, a seven-point self-monitoring test of the patients’ blood glucose (SMBG) was done at least once a week. A routine evaluation, including glycosylated hemoglobin (HbA1C), serum creatinine, SGPT, blood pressure, fundus examination, urine for the albumin creatinine ratio (ACR) and an ECG, was done at the baseline and, then, after six months.

Quantification of the qualitative intangible parameters was done utilizing questionnaires to evaluate the quality of life (Ferrans and Powers Quality of Life Index), Diabetic Distress Score (DDS) and day-to-day experiences by the Mindful Attention Awareness Scale (MAAS) (assessed only in the intervention group). All the questionnaires were linguistically adapted and pre-validated in 20 healthy subjects.

The Ferrans and Powers Quality of Life Index—Generic Version (QLI—G) is a 66-item inventory rated on a 6-point Likert-type scale. The questionnaire is split into two parts: the first section contains 33 questions relating to general satisfaction, and the remaining half contains a similar number of questions relating to values. Quality of life has proven to be a valid and reliable marker of patients’ health and well-being. Ferrans defines the quality of life as “a person’s sense of well-being that stems from satisfaction or dissatisfaction with the areas of life that are important to him/her”. The Ferrans and Powers Quality of Life Index—Generic Version (QLI—G) is a valuable tool in measuring this construct, with a large number of citations demonstrating its robustness [5].

The diabetes distress score (DDS) is a well-known questionnaire that has frequently been used in multiple studies [6]. The DDS has 4 subdomains: Emotional Burden (EB), Regimen-Related Distress (RRD), Physician-Related Distress (PRD) and Interpersonal Distress (IPD) [6].

The Mindful Attention Awareness Scale (MAAS) is a 15-item scale designed to assess a core characteristic of dispositional mindfulness—namely, the open or receptive awareness of and attention to what is taking place in the present. The scale shows strong psychometric properties and has been validated with college, community and cancer patient samples. Correlational, quasi-experimental and laboratory studies have shown that the MAAS taps a unique quality of consciousness that is related to, and predictive of, a variety of self-regulation and well-being constructs. The study takes 10 min or less to complete [7].

### Statistical Analysis

Data was entered and analyzed using the IBM Statistical Package for Social Sciences (SPSS version 25.0, SPSS, Inc., Chicago, IL, USA) for Macintosh. The continuous variables were expressed as the mean (SD) and median (range) and the categorical variables as the frequency with percentages. A paired *t*-test was used to calculate the differences in the glycemic control and quality of life before and after the intervention. A *p*-value of less than 0.05 was considered statistically significant.

## 3. Results

A total of thirty-two adults (15 males, 46.9%; mean age 23.8 ± 6.6 years) with type 1 diabetes (mean diabetes duration 12.7 ± 6.2 years) were enrolled in the study. The participants were equally randomized to the intervention and control groups. There were no statistical differences between the two groups in terms of the age, duration of diabetes, weight, body mass index (BMI) and the baseline laboratory parameters (Table 1). Hence, the two groups were statistically comparable for studying the effects of meditation on the glycemic control and quality of life. We noted that four patients (25%) in the intervention group and two (12.5%) in the control group had coexisting hypothyroidism. In addition to insulin therapy, one patient in each group was also receiving metformin for diabetes treatment.

### 3.1. Quality of Life and Diabetes Distress Scale (DDS) Score At the Baseline

A comparison of the baseline quality of life (QoL) and Diabetes Distress Scale (DDS) scores among both the groups is summarized in Table 2. No significant differences in the mean diabetes distress score and quality of life, except the family domain, were seen among the two groups.

### 3.2. Quality of Life and Diabetes Distress Scale (DDS) Score at 6 Months

At the end of the study period, we found a significant reduction in the total diabetes distress score in the intervention group (1.6 ± 0.3 versus 1.3 ± 0.3; *p* = 0.003), while no change was observed in the control group (1.6 ± 0.7 versus 1.7 ± 0.4; *p* = 0.762). A significant improvement in the emotional burden and regimen-related distress subscales was seen in the intervention group. No significant change was observed in the physician-related distress and interpersonal distress subscales. Among the four domains of the quality of life, a statistically significant improvement was noticed in the health and functioning domain in the intervention group (*p* = 0.023). No change was observed in the control group in terms of the four domains of the quality of life (Table 3).

### 3.3. Metabolic and Glycemic Control Before and After the Intervention

At the end of six months, a statistically significant improvement was seen in the mean blood glucose in the control group (222.4 ± 77.8 versus 182.6 ± 52.0; *p* = 0.007) and the intervention group (215.3 ± 50.1 versus 193.2 ± 31.8; *p* = 0.008). We also observed a statistically significant improvement in the blood pressure and HbA1C in both groups (Table 4).

### 3.4. Mindfulness Attention Awareness Scale before and after the Intervention

Additionally, we observed no change in the MAAS scale at the end of the study period in the intervention group (4.2 ± 0.6 versus 4.6 ± 0.9; *p* = 0.214) (Figure 1).

## 4. Discussion

We believe this to be the first Indian study to determine the impact of Mindfulness Meditation on the glycemic control and quality of life in people living with type 1 diabetes.

Although a remarkable improvement in glycemic control has been reported in previous studies among individuals living with T2D [8,9], there is a scarcity of literature on the efficacy of Mindfulness Meditation in people living with T1D [10]. We observed a significant reduction in the mean HBA1C level in both the control and the intervention groups. The improved the glycemic control in the control group could plausibly be explained by the improved self-monitoring of patients’ blood glucose levels, since both of the groups were instructed to maintain a record of a seven-point blood glucose test at least once a week.

Diabetes and its comorbidities are indeed associated with psychological distress [11]. In agreement with the previous literature [12], we also observed impaired quality of life and psychological distress among our patients, in both the control and intervention groups.

Further, it is well-known that stress leads to impaired glycemic control among diabetics [13]. While it may not always be possible to avoid stress, learning to cope with it certainly results in good glycemic control, as shown in a study by Kian et al. [9]. Reduced psychological stress and an improved quality of life have been reported following Mindfulness Meditation [10]. In accordance with the previous studies, in our study, the intervention group showed improvement in the quality of life (QoL), whereas, in the control group, no effect was observed. Similarly, in a study by Chung et al., a higher quality of life level (*p* < 0.001), anxiety reduction and blood pressure than the control but a similar anxiety level (*p* = 0.74) was reported [14].

We also observed a significant improvement in the blood pressure control in both groups. A similar reduction in the mean arterial pressure by 6mm Hg was reported by Rosenzweig et al. in their study involving 14 T2D participants [8]. An overall improvement in lifestyle could possibly explain the improved blood pressure in the control group. The fact that their sugar readings were being monitored on a regular basis most likely encouraged the participants in the control group to follow a more balanced lifestyle.

The small sample size was a limitation of the present study. Larger multi-centric, randomized controlled trials are needed to support this hypothesis.

To conclude, an improvement in the glycemic control and quality of life was observed in our study. Mindfulness Meditation certainly plays an important role in attaining peace of mind and helps patients to channel their energy in a positive direction. As clinicians, we should encourage our patients to practice mediation in their day-to-day life as an adjunctive therapy.

## Figures and Tables

**Figure 1 medsci-09-00033-f001:**
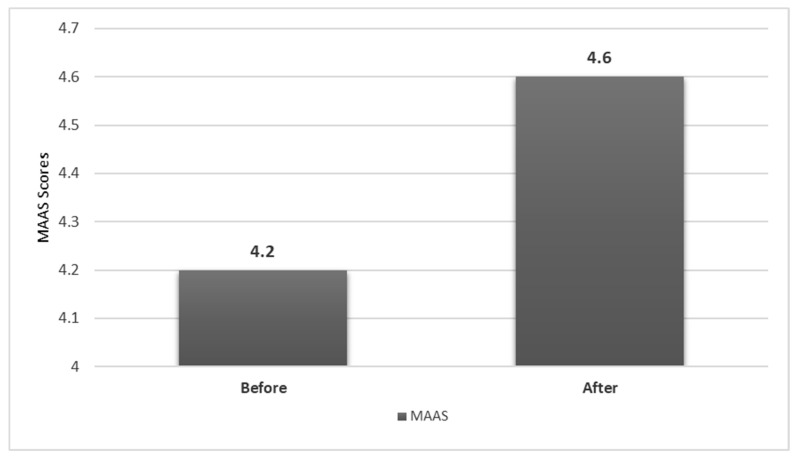
Bar diagram showing the change in the Mindful Attention Awareness Scale (MAAS) before and after the study in the intervention group (*n* = 16).

**Table 1 medsci-09-00033-t001:** Baseline characteristics of the participants in the control and intervention groups.

Variables	Control Group (*n* = 16)	Intervention Group (*n* = 16)	*p*-Value *
Age (years)	22.8 ± 3.9	24.8 ± 8.6	0.404
Duration of Diabetes (years)	12.2 ± 6.1	13.25 ± 6.44	0.634
Weight (kg)	55.8 ± 10.7	55.9 ± 11.4	0.972
Systolic BP (mm Hg)	131.1 ± 12.9	129.9 ± 10.4	0.788
Diastolic BP (mm Hg)	87.2 ± 9.5	88.7 ± 6.3	0.602
BMI (kg/m^2)^	21.5 ± 4.1	21.6 ± 4.4	0.972
Hemoglobin (g/dL)	12.8 ± 1.9	13.6 ± 1.4	0.199
Serum Creatinine (mg/dL)	0.73 ± 0.19	0.76 ± 0.19	0.561
SGPT (mg/dL)	19.9 ± 9.9	17.5 ± 7.3	0.426
TSH (mIU/L)	3.1 ± 1.6	9.3 ± 15.3	0.128
HbA1c (%)	9.3 ± 2.3	9.1 ± 1.5	0.756
Urine ACR (g/mg)	16.1 ± 54.9	3.6 ± 6.2	0.372

* *p*-Value < 0.05 was considered as significant.

**Table 2 medsci-09-00033-t002:** Comparison of the quality of life (QoL) and Diabetes Distress Scale (DDS) score among the control and intervention groups at the baseline.

Variables	Control Group (*n* = 16)	Intervention Group (*n* = 16)	*p*-Value *
**Diabetes Distress Scale**			
Economic Burden	1.7 ± 0.6	2.1 ± 0.8	0.141
Physician Related Distress	1.4 ± 0.9	0.9 ± 0.1	0.141
Regimen Related Distress	2.1 ± 1.0	1.9 ± 0.6	0.710
Interpersonal Distress	1.5 ± 0.8	1.4 ± 0.6	0.518
Total Diabetes Distress Score	1.6 ± 0.7	1.6 ± 0.3	0.970
**Quality of Life**			
Health & Functioning	5.9 ± 3.2	6.9 ± 3.5	0.416
Social and Economic	8.4 ± 3.8	9.5 ± 3.6	0.401
Psychological/Spiritual	7.6 ± 4.1	9.5 ± 4.2	0.200
Family	8.6 ± 3.9	11.4 ± 3.2	0.037 ***
Total QoL Score	6.5 ± 3.0	8.8 ± 3.4	0.054

* *p*-Value < 0.05 was considered as significant.

**Table 3 medsci-09-00033-t003:** Comparison of the quality of life (QoL) and Diabetes Distress Scale (DDS) score among the control and intervention groups after the intervention.

Variables		Before Intervention	After Intervention	*p*-Value *
**Diabetes Distress Scale**				
Economic Burden	Control	1.7 ± 0.6	2.1 ± 0.6	0.069
	Intervention	2.1 ± 0.8	1.5 ± 0.4	0.021 *
Physician Related Distress	Control	1.4 ± 0.9	1.1 ± 0.1	0.268
	Intervention	0.9 ± 0.1	1.1 ± 0.3	0.249
Regimen Related Distress	Control	2.1 ± 1.0	1.9 ± 0.6	0.574
	Intervention	1.9 ± 0.6	1.3 ± 0.4	0.001 *
Interpersonal Distress	Control	1.5 ± 0.8	1.6 ± 0.8	0.795
	Intervention	1.4 ± 0.6	1.2 ± 0.3	0.199
Total Diabetes Distress Score	Control	1.6 ± 0.7	1.7 ± 0.4	0.762
	Intervention	1.6 ± 0.3	1.3 ± 0.3	0.003 *
**Quality of Life**				
Health & Functioning	Control	5.9 ± 3.2	7.8 ± 3.1	0.121
	Intervention	6.8 ± 3.6	9.6 ± 3.3	0.023 ***
Social and Economic	Control	8.4 ± 3.8	6.6 ± 3.7	0.086
	Intervention	9.5 ± 3.6	10.7 ± 3.4	0.158
Psychological/Spiritual	Control	7.6 ± 4.1	7.9 ± 3.8	0.781
	Intervention	9.5 ± 4.2	11.2 ± 3.9	0.174
Family	Control	8.6 ± 3.9	8.9 ± 3.8	0.821
	Intervention	11.4 ± 3.2	11.4 ± 4.1	0.957
Total QoL Score	Control	6.6 ± 3.0	6.9 ± 3.1	0.670
	Intervention	8.8 ± 3.4	10.3 ± 3.0	0.162

* *p*-Value < 0.05 was considered as significant.

**Table 4 medsci-09-00033-t004:** Change in the body mass index, blood pressure and glycemic control before and after the intervention in the control and intervention groups.

Variables		Before Intervention	After Intervention	*p*-Value *
BMI (kg/m^2)^	Control	21.5 ± 4.1	21.7 ± 4.6	0.618
	Intervention	21.6 ± 4.4	21.9 ± 4.8	0.239
Systolic BP (mm Hg)	Control	131.1 ± 12.9	121.4 ± 11.3	0.004 *
	Intervention	129.9 ± 10.4	124.0 ± 14.4	0.064
Diastolic BP (mm Hg)	Control	87.2 ± 9.5	79.5 ± 5.9	0.003 *
	Intervention	88.7 ± 6.3	80.2 ± 8.6	0.002 *
HbA1c (%)	Control	9.3 ± 2.3	8.1 ± 1.5	0.005 *
	Intervention	9.1 ± 1.5	8.4 ± 0.9	0.008 *
Mean Blood Glucose (mg/dL)	Control	222.4 ± 77.8	182.6 ± 52.0	0.007 *
	Intervention	215.3 ± 50.1	193.2 ± 31.8	0.008 *

* *p*-Value < 0.05 was considered as significant.

## Data Availability

The data presented in this study are available on request from the corresponding author. The data are not publicly available due to the ethical and privacy restrictions.

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
