# Peer review of "Mindfulness Meditation as Adjunctive Therapy to Improve the Glycemic Care and Quality of Life in Patients with Type 1 Diabetes"

_medsci, 2021, doi:10.3390/medsci9020033_

Round 1
Reviewer 1 Report
In this randomized controlled study, the Authors studied the effect of mindfulness meditation on glycemic control and Quality of Life in people living with Type 1 Diabetes.
They found an improvement in the biochemical parameters in the intervention group as well as in control group. Moreover, the authors observed a significant reduction in the total diabetes distress score and in the in the health and functioning domain of the QoL assessment only in the intervention group.
Even if the topic of the paper could be of some interest for the reader, the paper has some issues that might to be addressed:
- Please provide an explanation about the randomization method.
- The discussion section appears not clear, that may pose some difficulties to readers when trying to understand which is the focus of the paper. Please carefully review this section by better explaining how the results have filled the gap that was identified in the introduction as well as providing caveats to the interpretation.
- It may be useful to divide results section by subheadings.
Reviewer 2 Report
In this study, the authors investigated the effects of mindfulness meditation on glycemic control in patients with T1D. A total of 32 T1D patients were randomly allocated to control or mindfulness mediation group. After follow up of 6 months, glycemic as well as blood pressure control were significantly improved in mindfulness meditation group.
Comments:
1) Did mindfulness mediation affect hypoglycemic episodes?
2) Did mindfulness meditation contribute to decrease daily insulin dose?
3) Did mindfulness meditation alter levels of insulin counter regulatory hormone?
Round 2
Reviewer 1 Report
No more comments, the manuscript has been
sufficiently improved to warrant publication in Medical Sciences